# Position: Trustworthy AI Agents Require the Integration of Large Language Models and Formal Methods

Yedi Zhang [* 1]   Yufan Cai [* 1]   Xinyue Zuo [* 1]   Xiaokun Luan [* 2]   Kailong Wang [* 3]   Zhe Hou [4]   Yifan Zhang [1]
Zhiyuan Wei [5]   Meng Sun [2]   Jun Sun [6]   Jing Sun [7]   Jin Song Dong [1]

## Abstract

Large Language Models (LLMs) have emerged as a transformative AI paradigm, profoundly influencing broad aspects of daily life. Despite their remarkable performance, LLMs exhibit a fundamental limitation: hallucination—the tendency to produce misleading outputs that appear plausible. This inherent unreliability poses significant risks, particularly in high-stakes domains where trustworthiness is essential. On the other hand, Formal Methods (FMs), which share foundations with symbolic AI, provide mathematically rigorous techniques for modeling, specifying, reasoning, and verifying the correctness of systems. These methods have been widely employed in mission-critical domains such as aerospace, defense, and cybersecurity. However, the broader adoption of FMs remains constrained by significant challenges, including steep learning curves, limited scalability, and difficulties in adapting to the dynamic requirements of daily applications. To build trustworthy AI agents, we argue that the integration of LLMs and FMs is necessary to overcome the limitations of both paradigms. LLMs offer adaptability and human-like reasoning but lack formal guarantees of correctness and reliability. FMs provide rigor but need enhanced accessibility and automation to support broader adoption from LLMs.

---

[*]Equal contribution   [1]Department of Computer Science, National University of Singapore, Singapore [2]School of Mathematical Science, Peking University, China [3]School of Cyber Science and Engineering, Huazhong University of Science and Technology, China [4]School of Information and Communication Technology, Griffith University, Australia [5]Beijing Institute of Technology, China [6]School of Computing and Information Systems, Singapore Management University, Singapore [7]School of Computer Science, University of Auckland, New Zealand. Correspondence to: Yedi Zhang <yd.zhang@nus.edu.sg>.

*Proceedings of the $42^{nd}$ International Conference on Machine Learning*, Vancouver, Canada. PMLR 267, 2025. Copyright 2025 by the author(s).

## 1. Introduction

The rapid advancement of modern AI techniques, particularly in the realm of Large Language Models (LLMs) like GPT (Achiam et al., 2023), Llama (Touvron et al., 2023), Claude (Claude, 2024), Gemini (Gemini, 2024), DeepSeek (DeepSeek-AI, 2025), etc., has marked a significant evolution in human-level computational capabilities. These models fundamentally reshape tasks across a spectrum of applications, from natural language processing to automated content generation. Trained on vast text corpora, LLMs excel in generating responses that are contextually accurate and stylistically appropriate. However, their applicability in safety-critical or knowledge-critical settings remains limited due to their inherent reliability issues—primarily, their propensity for generating outputs that, while plausible, may be factually incorrect (Jacovi & Goldberg, 2020; Wiegreffe & Marasovic, 2020; Agarwal et al., 2024). This limitation, known as "hallucination", stems from the probabilistic nature of learning-based AI, where the models optimize for likelihood rather than truth or logical consistency. Even worse, hallucination is mathematically proven inevitable for LLMs (Xu et al., 2024).

In contrast, Formal Methods (FMs) have been established as rigorous tools for verification and validation of critical systems where failure is intolerable, such as aerospace (relevant areas including avionics) (Dragomir et al., 2022; Liu et al., 2019), autonomous driving (König et al., 2024; Alves et al., 2021; Huang et al., 2022), and medical devices (Freitas et al., 2020; Arcaini et al., 2018). These methods are designed to ensure the correctness and safety of hardware and software systems by performing rigorous mathematical analysis. Despite the demonstrated benefits, the adoption of FMs remains limited, primarily due to their significant computational complexity and the specialized expertise required for the implementation.

Although both computational paradigms encounter inherent challenges of their own—namely, the unreliability stemming from the statistical nature of LLMs and the high barrier and complexity of FMs—recent studies have highlighted their potential for mutual benefits (Wu et al., 2022; Pan et al., 2023; He-Yueya et al., 2023; Zhou et al., 2024; Yang &

Deng, 2019; Yang et al., 2024; Song et al., 2024; Cai et al., 2025). Efforts to bridge these two paradigms aim to harness their respective strengths, with the ultimate objective of developing a neural-symbolic AI that seamlessly integrates LLMs and FMs into a unified solution. For instance, to enhance the reliability of LLMs, various approaches (Pan et al., 2023; Ma et al., 2024b) have incorporated SMT solvers to facilitate reasoning tasks guided by specification rules or reasoning models derived from LLMs' context. Conversely, within the formal methods community, there is a growing trend to leverage LLMs to enhance the functionality and usability of automated verification (Wu et al., 2024; Wen et al., 2024). Moreover, some existing research has also focused on trustworthiness in general AI systems (Dalrymple et al., 2024; Wing, 2021; Seshia et al., 2022; Zhang et al., 2024), particularly in domains like robotics and cyber-physical systems, primarily using traditional FMs like model checking and verification. They emphasize challenges such as world modeling, specification design, and verifier implementation.

This paper advocates for **the fusion of LLMs and FMs as a necessary approach for building the next generation of AI agents.** Different from previous work on investigating an unidirectional approach to apply FMs to general AI systems or vice versa, we give a bidirectional integration, highlighting how LLMs can enhance FMs to improve FMs' efficiency and adaptability, and how FMs can help certify LLM-driven agents' trustworthiness. Finally, by leveraging their complementary strengths, we propose a framework that enhances reliability, ensures provable correctness, and mitigates risks in AI-driven decision-making processes. Through case studies and conceptual explorations, we demonstrate how this integration can bridge neural learning and symbolic reasoning, ultimately fostering more trustworthy AI systems.

## 2. Alternative Views

**Relying Solely on Natural Language Reasoning.** Natural language reasoning (Yao et al., 2023) enables LLMs to process and generate information in an intuitive, human-like manner. However, it lacks rigorous correctness guarantees, making it unreliable for high-stakes decision-making. While statistical reasoning—areas where LLMs excel—can be more efficient than strict formalism in certain domains, the inherently learning-based nature, lacking rigorous reasoning, can lead to hallucinations and logical errors.

**Relying Solely on Expert Systems.** Traditional expert systems (Jackson, 1986) rely on fixed rule-based ontologies and usually operate under a closed-world assumption (CWA), i.e., anything not explicitly stated as true is assumed false. This rigid constraint is insufficient for open-ended, real-world reasoning, where knowledge is incomplete and context-dependent. Furthermore, expert systems struggle to adapt to new information or make inferences beyond

their predefined rules, making them inadequate for complex, evolving problem domains.

Instead of the rigid CWA, integrating open-world assumption techniques with formal constraints enables greater reasoning flexibility. This approach accommodates incomplete or evolving domain knowledge under uncertainty, thereby enhancing adaptability, statistical analysis, and uncertainty management—key strengths of modern AIs like LLMs.

**Relying Solely on LLMs.** LLMs, in their current form, are not inherently trustworthy for critical applications such as law, healthcare, and other safety-critical systems (Armstrong, 2023; Bellware & Masih, 2024; Choudhury & Chaudhry, 2024). LLMs exhibit hallucinations, lack of traceability, and non-deterministic behavior, rendering them unsuitable for scenarios requiring strong guarantees in terms of explainability, correctness, and security. While the integration of LLMs with Retrieval-Augmented Generation (RAG) (Lewis et al., 2020; Guu et al., 2020) aims to mitigate some of these limitations by providing access to external knowledge sources, this approach does not inherently improve reasoning capabilities (Chen et al., 2024b). Indeed, RAG-enhanced models primarily enhance factual accuracy by retrieving relevant documents but do not ensure logical coherence, consistency, or rigorous deductive reasoning. These models often face challenges in executing deep reasoning, a capability that cannot be easily and effectively achieved through mere fine-tuning (Liu et al., 2024).

Instead of rejecting formal reasoning outright, a more effective approach involves controlled augmentation, where LLMs are integrated with formal methods tools, such as proof-checkers and SMT solvers.

**Relying Solely on FMs.** Formal methods are mathematically rigorous methods that rely on manually defined specifications and inference rules for modeling, reasoning, and verifying the systems. However, their application in dynamic, real-world environments presents several challenges (Kneuper, 1997; Batra, 2013). FMs often struggle with scalability due to the computational demands of exhaustive state-space exploration, making them impractical for large-scale systems. Additionally, not all system aspects can be fully formalized, particularly in unpredictable environments, leading to gaps in formal analysis. The complexity and resource intensity of developing formal specifications and conducting proofs further limit their widespread adoption in the industry (Kaleeswaran et al., 2023).

A hybrid approach integrating adaptability, learning ability, and natural language-based heuristics can provide a practical middle ground. For instance, LLM-assisted theorem provers, where LLMs support automated reasoning by generating proofs or proposing logical constraints, can help bridge such gaps by retaining the flexibility of LLMs while mitigating

their weaknesses through formal guarantees.

## 3. LLM for FM: Verifying Intelligently

To achieve deep integration of LLMs and FMs for enhancing or certifying the trustworthiness of LLM agents, it is crucial to improve the scalability and automation capabilities of FMs, and LLMs offer promising solutions to these challenges. Accordingly, this section explores how LLMs can augment existing FMs by enabling the development of intelligent agents capable of performing complex tasks such as model checking and theorem proving. Indeed, FMs currently face significant obstacles to widespread industrial adoption, including the difficulty of formalizing requirement specifications, the limited scalability of algorithms in large-scale systems, and the considerable manual effort required for proof construction and validation. In contrast, LLM agents bring adaptability and efficiency to traditional formal verification processes, paving the way for more automated and effective formal methods. With their ability to process and generate structured code and symbolic representations, LLMs can serve as intelligent assistants in automating tedious tasks within formal methods workflows.

### 3.1. LLM for Autoformalization

Autoformalization is the process of automatically translating natural language-based specifications or informal representations into formal specifications or proofs. This complex task demands a deep understanding of both informal and formal languages, along with the ability to generate accurate, machine-readable formal representations. Recent research has demonstrated the effectiveness of LLMs in various auto-formalization scenarios, including neural theorem proving (Jiang et al., 2023), temporal logic generation (Murphy et al., 2024), and program specification generation based on source code (Ma et al., 2024a). In this section, we show the role of LLM agents in facilitating proof auto-formalization.

Informal proofs, commonly found in textbooks, research papers, online forums, or even generated by LLMs, often omit details that humans consider trivial or self-evident. However, to ensure rigorous verification by theorem provers, they need to be translated into formal proofs that adhere to a specific syntax, where all the details are explicitly provided. We provide one motivating example in Appendix A.

To address this, we propose using auto-formalization agents equipped with enhanced capabilities for symbolic reasoning. More specifically, auto-formalization agents break down the process into manageable steps: (i) generating proof outlines, (ii) filling intermediate steps using external tools, and (iii) integrating and refining proofs. To elaborate, the agent first constructs a high-level proof outline, capturing the main steps of the informal proof while leaving placeholders for missing intermediate steps. This outline aligns with the informal proof structure and serves as a blueprint for the subsequent formalization process. The agent delegates the task to external tools like computer algebra systems for the missing details, especially those involving symbolic reasoning or algebraic manipulations. These tools can perform accurate transformations on the mathematical expressions, ensuring the correctness of the derived intermediate steps. Once the intermediate steps are derived, the agent integrates them into the proof outline, filling in the placeholders and completing the formalization. If the agent still encounters gaps in specific steps, it iteratively refines the proof by revisiting the informal proof and consulting external tools. In this way, the auto-formalization agents can leverage the strong symbolic reasoning capabilities of external tools to fill in the missing details in the informal proofs, thus bridging the gap between informal and formal proofs and specifications.

### 3.2. LLM for Model Checking

Model checking is a formal verification technique that systematically explores the state space of a system to determine whether it satisfies specified properties, such as safety and liveness. It is particularly effective for finite-state systems, providing automated detection of logical errors like deadlocks or critical system property violations. However, traditional model checking faces great limitations, including scalability challenges for large systems and the complexity involved in system modeling and property formalization.

In this section, we illustrate how an LLM-enhanced model checking agent can address the aforementioned limitations by leveraging the strengths of LLMs. By integrating LLMs, such a model checking agent can accept system descriptions in natural language from users, generate corresponding formal models using the LLM, and iteratively refine these models based on feedback from the model checker. This synergy not only streamlines the model checking process but also enhances its accessibility for users without deep expertise in formal verification.

We demonstrate this insight through a model checking agent framework built around a widely adopted model checker, Process Analysis Toolkit (PAT) (Sun et al., 2008; Liu et al., 2011). PAT is a formal verification tool designed for modeling, simulating, and verifying concurrent and real-time systems. It supports the verification of key properties such as deadlock-freeness, reachability, and refinement, addressing critical correctness and reliability concerns in system design. Widely applied in domains such as automotive and aerospace, resource optimization, and complex system analysis, PAT provides a robust and well-established foundation for the development of a model checking agent.

**Example.** In car system development, preventing key lock-

in is essential for user convenience, avoiding costly locksmith services and severe delays. To ensure such incidents do not occur, the system must maintain logical consistency across operations.

We first prompt an LLM (gpt-4o-2024-08-06) to generate a formal model directly from a detailed description. While the model follows learned syntax rules and demonstrates planning capabilities, it contains a critical logic flaw: it allows the key to be locked inside the car. This issue stems from the hallucination of GPT-4o, incorrectly assuming that placing the key in a locked car is valid.

For example, the following transition allows the action *leavekey* to occur when the key is held by *owner i* and *owner i* is actually near the car:

```
[key == i && owner[i] == near]leavekey{key = incar;}
```

However, the action lacks a necessary condition: the car door must be open. By omitting this constraint, the model permits an invalid state where the key is left inside a locked vehicle. Consequently, the resulting system design deviates from the intended behavior, compromising its reliability.

To address this issue, we use PAT to formally verify the generated system. PAT detects an error trace, a sequence of operations that lead to the key being locked inside, revealing logical flaws in the key and door logic. By analyzing this trace, the LLM identifies the flaw and corrects it by imposing stricter restrictions on when the key can be left inside. It also defines clear conditions for locking the door, ensuring alignment with the intended system behavior.

```
[key == i && owner[i] == near && door == open]
leavekey{key = incar;}
```

```
[(owner[i] == near && key == i) || owner[i] == in]
lockdoor.i{door = lock;}
```

The refined model, incorporating PAT's feedback, ensures the key can never be locked inside the car and passes formal verification. This example demonstrates the powerful synergy between formal verification and LLM-driven development: LLMs streamline system development, while PAT ensures rigor by detecting and correcting logical inconsistencies. Together, they enable a robust, user-friendly approach to formal system development, ensuring that critical requirements are met with precision.

**Prototype.** To implement the PAT model checking agent, we design a custom pipeline that integrates LLMs with formal verification tools to support automated code generation, refinement, and validation. This setup enables iterative interaction between components while preserving precise control over each stage of the pipeline.

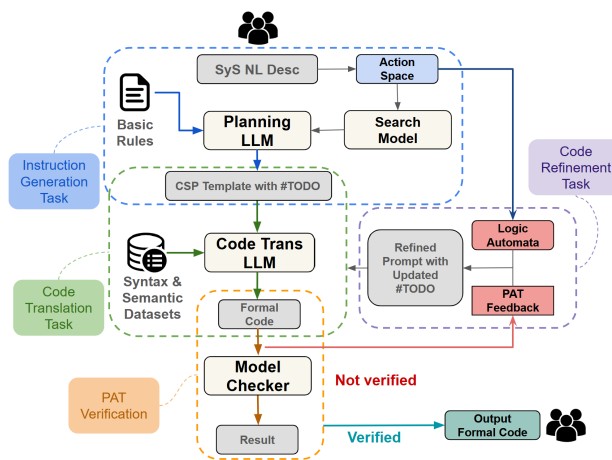

*Figure 1.* PAT agent prototype.

Figure 1 illustrates the PAT agent prototype, which follows a structured workflow. The process begins with the user providing a natural language description of a system, including its desired behaviors and properties, such as a mutual exclusion protocol and its expected safety and liveness conditions or behaviors.

An LLM with strong reasoning capabilities then processes the input to generate a structured implementation plan. This step defines an action space, employs a search model to identify feasible actions, and the LLM organizes them into a logical breakdown of the system. The plan includes precise mappings of logical steps, such as variable definitions, state transitions, and system properties, forming the foundation for NL-to-code translation.

Following the structured plan, a specialized LLM trained in syntax and logic generates the code and assertions needed to implement the system. Rather than generating code from scratch, the model treats this as an NL-to-code translation task, filling in details based on the structured plan. This approach enhances precision by breaking down code generation into distinct planning and translation steps, making the process more manageable.

The generated code and assertions are submitted to an automated verification tool, specifically PAT, to identify issues such as syntax errors and logical inconsistencies. If verification fails, a refined prompt is created by incorporating PAT feedback and comparing the implementation with the ideal automata outlined during planning. This iterative refinement continues until all properties are satisfied, enabling an automated yet rigorous approach to system development that enhances efficiency while ensuring correctness.

Unlike typical LLM applications that can tolerate ambiguity or loose semantics, our model checking agent enforces strict formal guarantees by ensuring that all generated out-

puts are provably correct through formal verification. The framework utilizes the strengths of LLMs, such as natural language understanding and code synthesis, within a rigorously structured setting that incorporates iterative correctness checking and refinement. By providing a user-friendly interface, the agent empowers users without formal methods expertise to generate verified system models, promoting the broader adoption of trustworthy and verifiable AI tools in system development.

**Generalization.** Since PAT is a low-resource formal language with limited training data, enabling LLMs to generate PAT models automatically presents a significant challenge. The strong performance of our framework on PAT therefore demonstrates its potential generalizability to other formal methods tools, such as Alloy Analyzer (Jackson, 2000), PRISM (Kwiatkowska et al., 2002), and UPPAAL (Behrmann et al., 2004). Its modular architecture allows the planning LLM to adapt to tool-specific semantics, while the Code Translation LLM generates corresponding formal code and assertions. By tailoring feedback and refinement loops to each tool, the framework supports seamless integration of LLM-driven development with a wide range of formal verification workflows.

### 3.3. LLM for Theorem Proving

Among all formal analysis techniques, theorem proving stands out for its capability to handle large state spaces, abstract specifications, and highly intricate systems. Unlike model checking, which is primarily designed for finite models and faces challenges with state space explosion, theorem proving excels in leveraging mathematical reasoning to establish properties that hold universally. This capability has been successfully demonstrated in critical systems, such as CompCert (Leroy, 2009), a formally verified C compiler that guarantees the correctness of compiled code, and seL4 (Klein et al., 2009), a microkernel with rigorous proofs of memory safety, functional correctness, and security properties. In this section, we explore how LLMs can enhance premise selection and proof generation for theorem proving and then illustrate the agent with one example in Appendix A.

#### 3.3.1. Premise Selection

Retrieving relevant facts from a large collection of lemmas is a critical task in theorem proving, a process known as premise selection. This task is typically done manually by explicitly specifying the used lemmas in the proof scripts, which often requires trial and error and deep domain knowledge, making it time-consuming and error-prone. Some powerful automation techniques in interactive theorem provers (ITPs) also need premise selection to first filter out irrelevant lemmas from the large search space. For exam-

ple, Sledgehammer (Böhme & Nipkow, 2010), an effective tool for Isabelle/HOL (Paulson, 1994), collects relevant facts from the background theories and sends them to external automatic theorem provers (ATPs) and SMT solvers to find proofs. This process involves premise selection to identify the most relevant lemmas that can help in proving the current goal, and Sledgehammer usually selects about 1,000 lemmas out of tens of thousands of available premises. Some heuristics (Meng & Paulson, 2009) and machine learning techniques (Kühlwein et al., 2013) like naive Bayes are used in Sledgehammer for relevant fact selection. Recent works (Mikuła et al., 2024; Yang et al., 2024) proposed using transformer models to learn the relevance of lemmas for premise selection, which improved the success rate of Sledgehammer.

Our insight is that LLM agents can further improve the premise selection process by leveraging their code understanding capabilities. Premise selection fundamentally differs from other tasks like code retrieval. LLMs, with their strong code comprehension capabilities, offer a way to address this gap. They can infer the meaning of a lemma from its name, definition, and contextual information, mimicking the reasoning process of a human expert. For instance, a human expert can intuitively assess whether a given lemma is likely to be helpful for a particular proof goal. However, the sheer number of lemmas in large proof libraries makes it impossible for experts to evaluate and rank all possible candidates manually. By contrast, LLM agents can efficiently scale this process. We can first collect definitions and contextual information of lemmas and ask LLM agents to generate semantic descriptions in natural language for each lemma, forming a knowledge base for premise selection. Then, given a proof goal, LLM agents comprehend the goal and generate a semantic representation, which is used to query the knowledge base for relevant lemmas.

#### 3.3.2. Proof Generation

Proof step generation is the central task in theorem proving, where the objective is to predict one or more proof steps to construct a valid proof for a given theorem. Many pioneering works on LLM-based proof generation (Polu & Sutskever, 2020; Polu et al., 2023; Han et al., 2022) approach this problem as a language modeling task and train LLMs on large-scale proof corpora to predict the next proof step. Various techniques have been developed to improve the quality of generated proofs. For instance, learning to invoke ATPs to discharge subgoals (Jiang et al., 2022), repairing failed proof steps by querying LLMs with the error message (First et al., 2023), and predicting auxiliary constructions to simplify proofs (Trinh et al., 2024) have all demonstrated significant potential.

However, real-world application scenarios present chal-

lenges that go beyond these methods. Human experts, for instance, do not solely rely on immediate proof context or predefined strategies. Instead, they first have a high-level proof plan in mind and need to consult the definitions of important concepts or theorems during the proof process frequently. Additionally, experts often employ a trial-and-error approach, iteratively refining their methods to construct a valid proof. This highlights a limitation of current LLMs when used as standalone tools: while they excel at producing plausible proof steps, they lack broader strategic reasoning and adaptability. This gap makes it difficult for LLMs to consistently surpass human performance in proof generation tasks.

To address these limitations, our insight is to propose a shift toward LLM agents that more closely emulate human experts in their proof strategies. In contrast to standalone LLMs, these agents integrate multiple capabilities, allowing them to reason, adapt, and interact during the proof generation process. This distinction can be articulated through the following two key features:

**Feature 1. Explicit Proof Intentions.** A defining feature of LLM agents is their ability to generate both proof steps and explicit proof intentions—statements that explain the reasoning or goals underlying each step. This additional layer of information is critical for improving both automated and human-driven refinement. When a proof step fails, the agent can use the intention, along with error feedback, to attempt a proof repair. Even if the repair is unsuccessful, the intention provides valuable insights for human users, streamlining their efforts to identify and resolve the issue.

**Feature 2. Dynamic Retrieval of Relevant Knowledge.** LLM agents go beyond the immediate context by incorporating mechanisms to retrieve definitions, lemmas, or related theorems from knowledge bases. This mimics how human experts consult reference materials during the proof process but with significantly greater efficiency and scale. By dynamically identifying and incorporating relevant information, the agent can address gaps in its internal knowledge, enabling it to construct proofs that require broad or specialized domain understanding.

### 3.4. Trade-offs: LLM-augmented and Conventional FMs

The trade-off between LLM-augmented FMs and traditional FMs primarily lies in the balance between adaptability/expressiveness and precision/rigor. FM-only systems are distinguished by their high precision, offering exact formal specifications, rigorous proofs, and logically sound reasoning, thereby ensuring strong guarantees of correctness. However, these systems often struggle with flexibility and expressiveness when confronted with ambiguous, incomplete, or imprecisely defined real-world problems. In contrast, LLM-augmented FM systems introduce greater adaptability and expressive power, enabling more flexible handling of complex and informal inputs. However, this comes at the potential cost of consistency and precision in generated outputs, which may require manual verification to ensure correctness. Importantly, we argue that human involvement in validating formal specifications remains an unavoidable aspect even in purely FM-based systems. The integration of LLMs does not introduce fundamentally new challenges but rather shifts the focus of effort—reducing the burden of manual model specification construction and enhancing automation—while preserving the core objective of ensuring correctness, rigor, and robust system design.

## 4. FM for LLM: Towards Reliability

In this section, we illustrate how FMs can enhance the reliability of LLMs. Specifically, we explore this integration direction from three perspectives: (i) trustworthy LLMs with symbolic solvers, (ii) LLM Testing based on logical reasoning, and (iii) LLM behavior analysis. We argue that these FM-based techniques make AI systems reliably secure, paving the way for developing trustworthy AI systems.

### 4.1. SMT Solvers for LLM

Satisfiability Modulo Theories (SMT) solvers are specialized tools designed to determine the satisfiability of logical formulas defined over some theories, such as arithmetic, bit-vectors, and arrays. They play a pivotal role in formal verification, program analysis, and automated reasoning, serving as essential components to ensure the correctness and reliability of complex software systems.

Recent studies (Deng et al., 2024; Pan et al., 2023; Wang et al., 2024; Ye et al., 2024) have explored the integration of SMT solvers to enhance the accuracy and reliability of LLMs in logic reasoning tasks. These solver-powered LLM agents operate by translating task descriptions into formal specifications, delegating reasoning tasks to specialized expert tools for precise analysis, and subsequently converting the outputs back into natural language.

We have identified three main challenges within this research line. Firstly, while LLMs are capable of generating logical constraints or SMT formulas, they often produce suboptimal or overly verbose constraints, which can place an additional computational burden on the solver. Secondly, the outputs of LLMs lack guarantees of correctness or logical consistency, potentially introducing subtle inaccuracies or ambiguities in the generated SMT constraints. It can lead to invalid results or solutions that are challenging to interpret. Lastly, LLMs often lack domain-specific knowledge and may struggle to generate outputs that conform to the precise formal syntax required by SMT solvers. Conse-

quently, they may generate formulas that are semantically sensible but syntactically invalid formulas, rendering them unprocessable by the solvers.

We present our insights and proposed strategies to address the three key challenges outlined above.

**Strategy 1. Multiple LLMs Debating.** To address the challenge of LLMs generating suboptimal or overly verbose constraints, a potential strategy is leveraging multiple LLMs in a collaborative or adversarial framework to critique, validate, and refine each other's outputs. In this approach, the system employs one or more LLMs to generate SMT code from natural language inputs, while other LLMs function as "critics", evaluating the generated code for logical consistency, syntactic correctness, and alignment with the problem description. By incorporating feedback loops among these models, the system can iteratively refine the outputs and reduce ambiguity inherent in natural language inputs.

**Strategy 2. Test Generation.** Test cases will be automatically generated to validate the correctness and consistency of the LLM-generated SMT code against the expected behavior. Fuzzing techniques may also be employed to generate adversarial inputs for testing. Additionally, mutation-based approaches can be applied to both the SMT code and the natural language descriptions, with two LLMs comparing the resulting solutions. The strategy helps check the consistency between the natural language description and the SMT code produced by the LLMs.

**Strategy 3. Self-correction.** Feedback from tests, critics, or the solver itself can be leveraged to iteratively refine the SMT code. Errors identified via solvers can be categorized into syntax issues (e.g., invalid SMT-LIB syntax), semantic misalignments (e.g., logical inconsistencies), or performance bottlenecks (e.g., slow or incomplete solver responses). Based on this feedback, an LLM can be employed to debug and regenerate problematic parts of the constraints, ensuring that the refinements are both targeted and context-aware. This iterative refinement process, coupled with validation through re-testing, facilitates the convergence of LLM-generated SMT codes toward correctness and rigorousness. An example is given in Appendix A.

### 4.2. Logical Reasoning for LLM Testing

LLM Testing (Zhong et al., 2024; Hendrycks et al., 2021; Huang et al., 2023; Zhou et al., 2023) is primarily focused on establishing a comprehensive benchmark to evaluate the overall performance of the models, ensuring that they fulfill specific assessment criteria, such as accuracy, coherence, fairness, and safety, in alignment with their intended applications. An emerging research focus in this area is testing hallucinations in LLMs, with recent studies proposing various methods for their detection, evaluation, and mitigation.

A common and straightforward method is to create comprehensive benchmarks specifically designed to assess LLM performance. However, these methods, which often rely on simplistic or semi-automated techniques such as string matching, manual validation, or cross-verification using another LLM, have significant shortcomings in automatically and effectively testing Fact-conflicting hallucinations (FCH) (Li et al., 2024). This is largely due to the lack of dedicated ground truth datasets and specific testing frameworks. We contend that unlike other types of hallucinations, which can be identified through checks for semantic consistency, FCH requires the verification of content's factual accuracy against external, authoritative knowledge sources or databases. Hence, it is crucial to automatically construct and update factual benchmarks, and automatically validate the LLM outputs based on that.

To this end, we propose to apply logical reasoning to design a reasoning-based test case generation method aimed at developing an extensive and extensible FCH testing framework. Such a testing framework leverages factual knowledge reasoning combined with metamorphic testing principles to ensure a robust FCH evaluation of LLM.

#### 4.2.1. FACTUAL KNOWLEDGE EXTRACTION

This process focuses on extracting essential factual information from input knowledge data in the form of fact triples, which are then suitable for logical reasoning. Existing knowledge databases (Bollacker et al., 2007; Auer et al., 2007; Suchanek et al., 2007; Miller, 1995) serve as valuable resources due to their extensive repositories of structured data derived from documents and web pages. This structured data forms the foundation for constructing and enriching factual knowledge, providing a robust basis for the test case framework.

The extraction process typically involves structuring facts as three-element predicates, $nm(s, o)$, where "$s$" (stands for $subject$) and "$o$" (stands for $object$) are entities, and "$nm$" denotes the predicate. This divide-and-conquer strategy extracts facts category by category, effectively organizing information across various domains. The extraction process iterates through predefined categories of entities and relations, employing a database querying function to retrieve all relevant fact triples for a given entity and predicate combination. This ensures comprehensive and systematic extraction of factual knowledge, creating a well-structured dataset for reasoning and testing.

#### 4.2.2. LOGICAL REASONING

This step focuses on deriving enriched information from previously extracted factual knowledge by employing logical reasoning techniques. The approach utilizes a logical programming-based processor to automatically generate

new fact triples by applying predefined inference rules, taking one or more input triples and producing derived outputs.

In particular, to introduce variability in the generation of test cases, reasoning rules, commonly utilized in existing literature (Zhou et al., 2019; Liang et al., 2022; Abboud et al., 2020) for knowledge reasoning, are typically adopted, including negation, symmetric, inverse, transitive and composite. These rules provide a systematic framework for generating new factual knowledge, ensuring diverse and comprehensive test case preparation. The system applies all relevant reasoning rules exhaustively to the appropriate fact triples, enabling the automated enrichment of the knowledge base for further testing purposes.

### 4.2.3. BENCHMARK CONSTRUCTION

This process consists of two key steps: (i) generating question-answer (Q&A) pairs and (ii) creating prompts from derived triples, which together can significantly reduce manual effort in test oracle generation.

The question generation step uses entity-relation mappings to populate predefined Q&A templates, aligning relation types with corresponding question structures based on the grammatical and semantic characteristics of predicates. For predicates with unique characteristics, customized templates are employed to generate valid Q&A pairs. To enhance natural language formulation, LLM can be used to refine the Q&A structures. Answers are derived directly from factual triples, with true/false judgments determined by the data. Mutated templates, leveraging synonyms or antonyms, diversify questions with opposite semantics, yielding complementary answers. Then, prompts are designed in the second step to instruct LLMs to provide explicit judgments (e.g., yes/no/I don't know) and outline their reasoning in standardized formats. LLM analysts can utilize predefined instructions to ensure clarity and enable LLMs to deliver assessable and logically consistent responses. This method maximizes the model's reasoning capabilities within the structured framework of prompts and cues.

### 4.2.4. RESPONSE EVALUATION.

This step aims to enhance the factual evaluation in LLM outputs by identifying discrepancies between LLM responses and the verified ground truth in Q&A pairs. The key insight lies in constructing a similarity-based metamorphic testing and oracles to evaluate consistency by comparing the semantic structures of the response and ground truth, focusing on node similarity (fact correctness) and edge similarity (reasoning correctness). Responses are categorized into four classes: correct responses (both nodes and edges are similar), hallucinations from erroneous inference (nodes are similar, edges are not), hallucinations from erroneous knowledge (edges are similar, nodes are not), and overlaps

with both issues (both nodes and edges are dissimilar).

### 4.3. Rigorous LLM Behavior Analysis

While LLM testing techniques can effectively provide broad assessments and reveal edge cases that may provoke unexpected responses, they are limited in their capability to give rigorous guarantees on LLM behaviors. LLM verification, on the other hand, serves as a complementary mechanism. However, as LLMs grow more complex and tasks become increasingly sophisticated, traditional neural network verifiers lose relevance due to their limitations in accommodating diverse model architectures and their focus on single-application scenarios. Indeed, formal verification of LLMs poses intrinsic challenges due to three key factors:

**Factor 1. Non-Deterministic Responses.** Responses from LLMs are non-deterministic, meaning their outputs may vary even with the same input. This inherent variability presents substantial challenges to providing deterministic guarantees regarding their behavior.

**Factor 2. High Input Dimensions.** The high dimensionality of inputs in LLMs leads to exponential growth in the number of input tokens, rendering exhaustive verification across an infinite input space highly impractical.

**Factor 3. Lack of Formal Specification.** While formal specifications are rigorous, they often lack the expressive capability of natural language, which makes it extremely difficult to precisely capture the nuanced and complex language behaviors expected from LLMs. Hence, we propose that a specialized verification paradigm tailored specifically for LLMs should be considered to ensure reliable and rigorous certification for long-term applications.

Given these challenges, we argue that monitoring might serve as a viable long-term solution for reliable LLM behavior analysis. Positioned between testing and verification, monitoring of formalized properties at runtime enables rigorous certification of system behavior with minimal computation overhead by examining execution traces against predefined properties. This approach has already inspired several efforts to monitor LLM responses at runtime (Cohen et al., 2023; Manakul et al., 2023; Besta et al., 2024; Chen et al., 2024a). However, the specifications used in these methods remain ambiguous and informal. For example, they define the properties of low hallucination based on the stability of LLM outputs. More recently, an approach (Cheng et al., 2024) has been introduced to monitor the conditional fairness properties of LLM responses. The specifications in (Cheng et al., 2024) are informed by linear temporal logic and its bounded metric interval temporal logic variant, reflecting a shift toward formal methods for more precise and dependable monitoring of LLM behavior.

Despite these advancements, challenges remain in extending

such formal monitoring techniques to a broader spectrum of properties, including but not limited to robustness, factual consistency, adherence to ethical guidelines, and sensitivity to adversarial prompts. Real-world applications of LLMs often involve nuanced, context-dependent interactions that demand adaptive and scalable monitoring solutions. Future research should focus on integrating diverse monitoring approaches, incorporating statistical and formal analysis techniques with data-driven approaches to enhance adaptability, and leveraging real-time anomaly detection to enhance the comprehensiveness and practicality of LLM behavior monitoring in varied deployment scenarios, ultimately fostering greater trustworthiness and accountability in AI systems.

### 4.4. Trade-offs: FM-augmented and LLM-only Systems

A key advantage of hybrid LLM-FM systems lies in their ability to combine the generative flexibility of LLMs with the precision, rigor, and correctness guarantees provided by formal methods. However, these benefits come at a cost: increased system complexity and challenges in achieving seamless integration, particularly due to mismatches in representational languages and reasoning paradigms. Moreover, hybrid approaches may still be limited by the expressive capabilities of the underlying formal frameworks. Consequently, we argue that such integrations are most beneficial in downstream tasks where correctness, logical consistency, and verifiability are essential. However, for domains that prioritize creativity and expressive freedom over strict correctness, such as creative writing, LLM-only systems may remain the more suitable choice, given their emphasis on generative fluency and contextual adaptability.

## 5. Unifying Multiple FMs and LLMs

This section highlights the synergistic collaboration among multiple LLM and FM agents. By integrating the adaptability and expressiveness of LLMs with the rigorous formal guarantees of FMs, the hybrid approach enables the generation of reliable, verifiable, and logically sound system behaviors. The pipeline given in Figure 2 demonstrates how multiple FMs and LLMs are integrated to produce trustworthy actions. The process initiates with user-defined specifications or requirements, typically expressed in natural language. These inputs are first processed by an LLM specialized in auto-formalization, which translates informal natural language descriptions into structured formal representations. The resulting artifacts are then subjected to rigorous analysis through a pipeline of FM-based components, ensuring logical consistency and correctness with respect to the specified requirements. This hybrid workflow comprises the following six stages: (i) **Auto-formalization** enables LLMs to interpret natural language requirements and translate them into formal specifications; (ii) **Model checking**

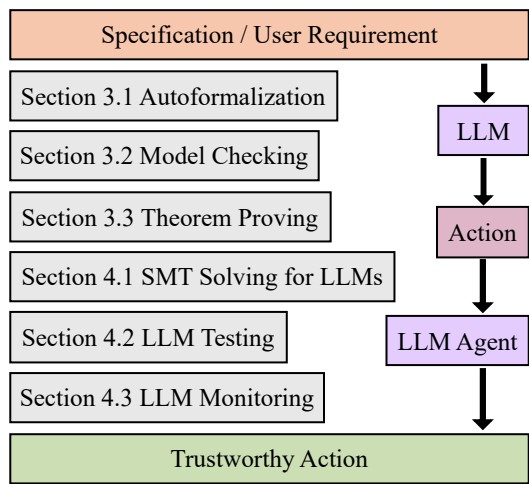

*Figure 2.* The framework of multiple LLM-FM agents.

verifies that the formalized specifications satisfy logical constraints, invariants, and safety properties; (iii) **Theorem proving** provides formal proof guarantees for critical properties by constructing machine-checkable proofs; (iv) **SMT solving** allows LLMs to generate formal encodings and leverage domain-specific SMT solvers to check logical satisfiability and uncover inconsistencies; (v) **LLM testing** complements formal analysis by identifying edge cases or emergent behaviors not covered by proofs; (vi) **LLM monitoring** monitors runtime behavior to ensure observed executions conform to expected outcomes and system-level specifications.

Across these stages, LLMs serve as intelligent intermediaries, integrating results, interpreting verification outcomes, and iteratively refining the system outputs. Moreover, the LLM agent adapts dynamically, modifying intermediate representations or generated actions in response to feedback from the verification pipeline; consequently, the final result manifests as a *trustworthy action*, aligning with the original user intent and satisfying formal correctness guarantees. By unifying multiple LLMs and FMs, the framework leverages their complementary strengths to enable systems that are not only adaptable and intelligent but also robust, interpretable, and trustworthy. More examples are given in Appendix A.

## 6. Conclusion

This paper advocates for the integration of Large Language Models and Formal Methods as a necessary approach to building trustworthy AI agents. Through case studies and conceptual explorations, we illustrate how this integration can address the inherent limitations of both paradigms. This fusion lays the foundation for bridging neural learning and symbolic reasoning, ensuring AI agents are both powerful and verifiably trustworthy.

## Acknowledgements

This study was supported by the Ministry of Education, Singapore under its Academic Research Fund Tier 3 (MOET32020-0003), the Ministry of Education, Singapore under its Academic Research Fund Tier 3 (Award ID: MOET32020-0004), the National Key R&D Program of China under Grant 2022YFB2702200, and the NSFC under Grant 62172019.

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

# A. Appendix

### A.1. Theorem Prover Agent

In this subsection, we describe the integration of an LLM agent with the Coq proof assistant. Coq is an interactive theorem prover that allows for the expression of mathematical assertions, formal verification, and the construction of proofs within a rigorous framework. By integrating Coq with an LLM agent, we aim to enhance the agent's ability to assist in formal proofs, reason about mathematical statements, and verify the correctness of solutions within the realm of formal logic.

**Overview of Coq Theorem Prover**    Coq is a proof assistant based on constructive type theory, which supports both functional programming and formal specification. Coq provides a framework for defining mathematical structures, functions, and proofs, leveraging a powerful type system to ensure correctness. It allows users to interactively develop proofs, and once a proof is verified, Coq guarantees its correctness by construction. Coq is widely used in formal verification, certified software development, and mathematical proof exploration. Integrating Coq with an LLM agent enhances the accessibility of formal proof construction and verification, allowing users to interact with formal methods in a more intuitive manner. This integration enables non-experts to explore and validate mathematical proofs without needing extensive familiarity with formal languages. Furthermore, the LLM agent can assist in automating proof steps, suggesting possible tactics, and generating human-readable explanations.

In the future, more advanced natural language translation mechanisms can be developed to handle increasingly complex theorems and mathematical domains. Additionally, the integration of other theorem provers with complementary strengths, such as Isabelle/HOL or Lean, can further broaden the agent's capabilities in formal reasoning and proof verification.

### A.1.1. CASE STUDY ON LLMS WITH COQ

To illustrate our perspective, we illustrate our recent exploration of the interaction between LLMs and Coq. Coq (Huet et al., 1997) is a classic proof assistant based on constructive type theory, supporting functional programming and formal specification. The integration of Coq with an LLM agent involves several key steps:

**Step 1. Natural Language Understanding.** The LLM agent receives natural language input from the user, typically in the form of a mathematical theorem, conjecture, or problem.

**Step 2. Formalization of the Problem.** The LLM agent translates the natural language problem into Coq's formal language. This includes defining types, propositions, and functions necessary for the formulation of the theorem.

**Step 3. Proof Construction.** The LLM agent collaborates with Coq to construct proofs, utilizing Coq's interactive features to propose proof steps that are subsequently verified or refined.

**Step 4. Proof Verification and Feedback.** Once the proof is constructed, Coq verifies its correctness. The LLM agent translates the verified proof back into natural language, providing a comprehensible explanation of the result to the user.

**Example**    Consider the following natural language query posed to the LLM agent:

*"Prove that the sum of two even numbers is even."*

The LLM agent first translates this into a formal Coq language:

```
Theorem sum_of_even_numbers : forall n m : nat,
  even n -> even m -> even (n + m).
```

The user needs to check the correctness of the coq code to guarantee consistency between the natural language and coq code. Then, the LLM agent will search the related theorems and lemmas, and construct the proof as follows:

```
Proof.
  intros n m Hn Hm.
  unfold even in *.
  destruct Hn as [k Hk].
  destruct Hm as [l Hl].
```

---

**Problem:** Let $u_n$ a sequence defined by $u_0 = 0$ and $\forall n \geq 0, u_{n+1} = 2u_n + (n+1)$. Show that $\forall n \geq 0, u_n = 2^{n+1} - (n+2)$.

**Informal Proof:** The property is true for $n = 0$, since $2^{0+1} - (0+2) = 0$. By induction, assuming the property holds for $n \geq 0$, we have: $u_{n+1} = 2u_n + (n+1) = 2\left(2^{n+1} - (n+2)\right) + n + 1 = 2^{n+2} - (n+3)$, which shows the property holds for $n+1$.

---

*Figure 3.* An informal proof example that needs to be formalized.

```
    exists (k + l).
    rewrite Hk, Hl.
    ring.
 Qed.
```

Coq verifies this proof, and the LLM agent explains and responds in natural language:

> *"The proof shows that if two numbers are even, their sum is also even by using basic properties of addition and the definition of even numbers."*

### A.2. An Motivating Example for Autoformalization

Specifically, Figure 3 gives a motivating example from the miniF2F benchmark (Zheng et al., 2022). The problem and its human-written informal proof are given, and it demonstrates a concise and correct inductive proof. However, some intermediate steps, particularly the algebraic manipulations, are omitted in the informal proof. Although the equation can be easily verified by human experts, it is challenging for LLMs to infer the missing details and generate a correct formal proof due to their limited symbolic reasoning capabilities. The following Isabelle/HOL proof snippet illustrates an attempt by ChatGPT-3.5-turbo to formalize the given informal proof. While the model correctly identifies the required transformations, it fails during the simplification of the term $2 * (n+2) - (n+1)$, mistakenly reducing it to $2n + 3$ instead of $n + 3$. This example highlights the limitations of LLMs in handling detailed algebraic reasoning without explicit guidance.

```
lemma induction_seq_mul2pnp1_forn:
  fixes n :: nat and u :: "nat ⇒ nat"
  assumes h0 : "u 0 = 0"
    and h1 : "⋀(n::nat). u (n+1) = 2 * u n + (n+1)"
    and h2 : "u n = 2 ^ (n+1) - (n+2)"
  shows "u (n+1) = 2 ^ (n+2) - (n+3)"
proof -
  have "u (n+1) = 2 * u n + (n+1)" using h1 by simp
  also have "... = 2 * (2 ^ (n+1) - (n+2)) + (n+1)" using h2 by simp
  also have "... = 2 * 2 ^ (n+1) - (2 * (n+2)) + (n+1)"
    by (simp add: right_diff_distrib')
  also have "... = 2 ^ (n+2) - (2 * (n+2) - (n+1))"
    sledgehammer
  (* an error in simplification *)
  also have "...  = 2 ^ (n+2) - (2*n + 3)" by auto
  also have "... = 2 ^ (n+2) - (n+3)" by auto
  finally show ?thesis by blast
qed
```

### A.3. Z3 Agent

To illustrate our perspective, we give our recent exploration of the interaction between LLMs and Z3 in Python.

Z3 (de Moura & Bjørner, 2008), a widely used SMT solver, accepts inputs in the form of simple-sorted formulas expressed in first-order logic (FOL). These formulas can include symbols with predefined meanings, defined by the underlying theories supported by the solver, and these theories encompass domains such as arithmetic, bit-vectors, arrays, etc., making Z3 particularly well-suited for reasoning about a wide range of logical constraints.

**Example** Consider a scenario where a user requests the LLM agent to solve a scheduling problem:

> *"Can you help plan a meeting for a team of three people—David, Emma, and Alex? David is free on Monday or Tuesday, Emma is free on Tuesday or Wednesday, and Alex is free only on Tuesday or Thursday. Find a common day when all three are available."*

We now provide a detailed, step-by-step solution for this task:

**Formalization of Constraints**    Given the above problem, the initial Z3 constraints in Python generated by the LLM are given as follows:

```python
# Define days of the week
days = ["Monday", "Tuesday", "Wednesday", "Thursday"]
David_free = [Bool(f"David_free_{day}") for day in days]
Emma_free = [Bool(f"Emma_free_{day}") for day in days]
Alex_free = [Bool(f"Alex_free_{day}") for day in days]
# Create a solver
solver = Solver()
# Define constraints for each person's availability
solver.add(Or(David_free[0], David_free[1]))
solver.add(Or(Emma_free[1], Emma_free[2]))
solver.add(Or(Alex_free[1], Alex_free[3]))
# Add constraints that ensure a common day for all three
common_day_constraints = [And(David_free[i],
Emma_free[i], Alex_free[i]) for i in range(len(days))]
solver.add(Or(common_day_constraints))
```

**Self-correction**    If the Z3 code has issues (e.g., missing constraints or syntax errors) or generates inconsistent results with the natural language description, the self-correction procedure will identify and correct them. In this example, the previous Z3 code ignores the following constraints:

```python
    # Constraints for David
    solver.add(And(Not(David_free[2]), Not(David_free[3])))
    # Constraints for Emma
    solver.add(And(Not(Emma_free[0]), Not(Emma_free[3])))
    # Constraints for Alex
    solver.add(And(Not(Alex_free[0]), Not(Alex_free[2])))
```

**Test Generation**    The agent mutates the constraints and tweaks the availability of each individual to create new conditions. For example, the new mutated constraints are David will be free on Monday and Wednesday. Emma will be free on Tuesday and Thursday. Alex will be free on Monday and Thursday. The updated Z3 code generated by the LLM is as follows:

```python
    # Mutated constraints for David
    solver.add(And(David_free[0], David_free[2]))
    solver.add(And(Not(David_free[1]), Not(David_free[3])))
    # Mutated constraints for Emma
    solver.add(And(Emma_free[1], Emma_free[3]))
    solver.add(And(Not(Emma_free[0]), Not(Emma_free[2])))
    # Mutated constraints for Alex
    solver.add(And(Alex_free[0], Alex_free[3]))
    solver.add(And(Not(Alex_free[1]), Not(Alex_free[2])))
```

The agent systems will check the consistency between the results produced by Z3 and the reasoning derived from natural language descriptions to further ensure the correctness of the Z3 codes.

**Multiple LLM Debating**    Whenever it comes to a collision between the Z3 reasoning results and the natural language reasoning results, the LLM debating will be activated to debate which part is correct. For example, after LLM-A generates the initial constraints and gets the results of Z3 code. LLM-B will critique the constraints, identifying potential issues such as missing exclusivity rules or improperly translated logic. LLM-C can suggest refinements, such as introducing mutual exclusivity or expanding constraints to handle edge cases. The consensus will be the output with the highest confidence score (e.g., most accurate or simplest) is selected for testing and execution.

**Problem Solving**    The translated constraints are fed into the Z3 solver, which checks the satisfiability of the formula and computes a solution if possible.

```python
    # Check for a solution
if solver.check() == sat:
    model = solver.model()
    common_days = [days[i] for i in range(len(days))
    if model.evaluate(David_free[i])
        and model.evaluate(Emma_free[i])
        and model.evaluate(Alex_free[i])]
    print(f"Common day(s) when everyone is free:
    {common_days}")
else:
    print("No common day when everyone is free.")
```

**Solution Interpretation**    The LLM agent receives the solution from the Z3 solver and translates it back into natural language for the user. The only day when all three are free is Tuesday. The output will be: Common day(s) when everyone is free: ['Tuesday'].

### A.4. Program Verification Example

Program verification is the process of ensuring that a program conforms to a formally defined specification. It involves the use of formal methods such as model checking, static analysis, and theorem proving to verify that the program behaves as intended. This process often requires specifying preconditions, postconditions, invariants, and loop variants to formally define the program's behavior. Tools such as Dafny, Why3, Frama-C, and SPARK provide automated and semi-automated support for verifying program properties.

The integration of program verification tools with an LLM agent has significant potential to make formal methods more accessible to a wider audience. The LLM agent can bridge the gap between natural language descriptions of program behavior and the formal specifications required for verification, thus enabling non-expert users to verify the correctness of their code. Additionally, the LLM agent can assist in identifying and correcting verification failures by providing meaningful explanations and suggesting potential fixes.

Future work may focus on enhancing the LLM agent's ability to handle more complex verification tasks, such as concurrent or distributed systems. Additionally, integrating multiple verification tools could provide more comprehensive verification capabilities, covering a broader range of programming languages and paradigms.

### A.4.1. LLM AGENT INTEGRATION

The integration of an LLM agent with program verification tools can be broken down into several stages:

1. **Natural Language Specification:** The LLM agent allows the user to describe program specifications in natural language. This includes stating what the program is supposed to do (e.g., sorting a list, finding the maximum value) and any specific requirements (e.g., ensuring the list is sorted in ascending order).

2. **Translation to Formal Specifications:** The LLM agent interprets the natural language specification and translates it into formal specifications, such as preconditions, postconditions, and loop invariants, using a formal specification language supported by the verification tool (e.g., ACSL for Frama-C, Boogie for Dafny).

3. **Program Analysis and Verification:** The program code and its formal specification are passed to a verification tool, which attempts to prove that the code adheres to the specification. The verification tool may automatically generate proofs, use SMT solvers, or require human-guided proof tactics.

4. **Feedback and Explanation:** Once verification is complete, the LLM agent presents the results to the user in natural language, explaining whether the program meets the specification and highlighting any verification failures or issues that need attention.

### A.4.2. MOTIVATING EXAMPLE

To illustrate the distinction between formal specification and executable implementation, we consider the problem of verifying whether a given array is a palindrome.

**Formal Specification**   We begin with a high-level, declarative specification of the palindrome property. This specification captures the essential symmetry of palindromes using a pure function over immutable sequences:

```
function method IsPalindrome(a: seq<int>): bool
  decreases a
{
  forall i :: 0 <= i < |a| / 2 ==> a[i] == a[|a| - i - 1]
}
```

*Listing 1.* Specification: Palindrome predicate.

This ghost function expresses the intended property mathematically: for all positions $i$ in the first half of the sequence, the element at position $i$ must equal the element at the symmetric position from the end. This abstract description is concise and easy to reason about but cannot be executed on its own.

**Verified Implementation**   To bridge the gap between high-level specification and executable code, we provide a concrete implementation over mutable arrays. The implementation uses a two-pointer technique and includes loop invariants to ensure correctness:

```
method CheckPalindrome(arr: array<int>) returns (res: bool)
  requires arr != null
  ensures res == IsPalindrome(arr[..])
{
  var left := 0;
  var right := arr.Length - 1;
  while left < right
    invariant 0 <= left <= right + 1 <= arr.Length
    invariant res == true
    invariant forall i :: 0 <= i < left ==> arr[i] == arr[arr.Length - i - 1]
    invariant forall j :: right < j < arr.Length ==> arr[j] == arr[arr.Length - j - 1]
  {
    if arr[left] != arr[right] {
      return false;
    }
    left := left + 1;
    right := right - 1;
  }
  return true;
}
```

*Listing 2.* Implementation: Palindrome check with invariants.

The example demonstrates the complementary roles of specification and implementation: the former provides a clear and mathematically grounded correctness criterion, while the latter ensures executable fidelity to that criterion through formal verification. Such separation of concerns is central to the design of trustworthy, formally verified systems.

