# OpenReview forum: "Position: Trustworthy AI Agents Require the Integration of Large Language Models and Formal Methods"
_ICML.cc/2025/Position_Paper_Track — ICML 2025 Position Paper Track poster_

### Official Review · Reviewer_JFYR · 2025-02-27

**Significance:** 3
**Argument Clarity:** 3
**Rating:** 4
**Confidence:** 3

**Questions:**

Please see the couple of weaknesses mentioned above.

**Discussion Potential:**

3

**Paper Summary:**

The paper argues that, to obtain trustworthiness in AI agents, integration of formal methods (FMs) with LLMs is necessary, giving the formal guarantees needed to underpin the creativity and adaptability afforded by the LLM. Conversely, they argue that LLMs can overcome some of the limitations of FM systems, enhancing such rigorous approaches with increased accessibility and automation.

Case studies from each perspective are covered to illustrate the potential benefits.

**Position:**

Yes

**Position In Title:**

Yes

**Related Work:**

3

**Strengths And Weaknesses:**

The paper is well written and makes its point clearly. The case studies are thorough, and provide a lot of background motivation and support for the argument. The issue of lack of trust in the output of LLMs is clearly of great importance.

No significant weaknesses, except maybe the following (and this is why I nudged the Support score down a notch):
- concerns re reliance on formalisation of natural language at various points (e.g. for theorem statements/sketches or for informal model specifications). This formalisation of course needs to be validated in order to provide the rigorous guarantees being sought. This bootstrapping problem is non-trivial and itself a major challenge.
- the example in A.4.1 is a bit trivial, e.g. the "ensure" clause in the code matches the code exactly! I would have thought an example where the proof of the post-condition was not trivial would be more illuminating/realistic.

**Support:**

2

---

> ### Author Rebuttal · Authors · 2025-04-01
>
> We sincerely appreciate the reviewer’s careful review and the time and effort invested in engaging with our paper. In the following, we provide our responses to the concerns raised by the reviewer.
>
> ### **Weakness**
>
> **W-1.**
>
> Thank you for raising the important concerns regarding the reliance on the formalization of natural language, particularly in the context of its validation for the rigorous guarantee being sought. We fully agree with the reviewer that guaranteeing the correctness of the formalization is indeed a non-trivial challenge. In response, we emphasize that ensuring its correctness necessitates human involvement, particularly during the modeling phase. It is important to note that achieving 100% correctness in modeling through automated methods remains unrealistic (given the intrinsic expressiveness gap between formal language and natural language). Nevertheless, automation can certainly reduce the labor-intensive aspects of the specification building or modelling process, though human oversight remains crucial for validation. Note that in a conventional FM-only system, the modeling and specification construction process is entirely manual, requiring human effort from start to finish.
>
> Indeed, much like code generation, where imperfections are recognized but the approach remains valuable due to the substantial reduction in manual effort, the same principle applies here. While achieving 100% correctness with the current state of technology is not feasible, automating parts of the formalization and verification processes still provides significant benefits. With human oversight, this approach can reduce human efforts and enhance efficiency while maintaining the integrity and reliability of the system.
>
> **W-2.**
>
> Thank you for pointing that out (we assume the reviewer is referring to A.4.2). We will include the following new example in A.4.2 to illustrate our position better.
>
> LLM formalizes the *Factorial problem* in Dafny with the following code:
>
> method Factorial(n: nat) returns (res: nat)
>
> &nbsp;&nbsp; requires n >= 0
>
> &nbsp;&nbsp; ensures res == (* i: nat {:trigger i} :: 1 <= i <= n ==> I)
>
> {
>
>   &nbsp;&nbsp; if n == 0 then return 1;
>
>   &nbsp;&nbsp; return n * Factorial(n - 1);
>
> }

---

> > ### Comment · Reviewer_JFYR · 2025-04-02
> >
> > Thank you for your responses to my review. I acknowledge your comments re W-1, and the proposed new example for W-2 would be acceptable. My rating of the paper remains unchanged.

---

### Official Review · Reviewer_YjPn · 2025-03-10

**Significance:** 4
**Argument Clarity:** 2
**Rating:** 2
**Confidence:** 4

**Questions:**

See the Strengths and Weaknesses section.

**Discussion Potential:**

2

**Paper Summary:**

This manuscript summarizes how LLMs and Formal Methods (FMs) can enhance each other’s capabilities. In terms of how LLMs can enhance FMs, this manuscript covers important aspects such as autoformalization (LLMs translating natural-language inputs into formal specification), assisting in model construction and checking, and serving as heuristics for various sub-problems. On the other hand, the manuscript also discusses how FMs can enhance the robustness or reliability of LLM-centric systems. For one, symbolic solvers can be leveraged by LLMs as external tools. In addition, symbolic verifiers can either facilitate LLM decoding (e.g., by providing verification feedback, facilitating prompt construction etc.) or build the foundation of LLM evaluations.



## update after rebuttal

I thank the authors for their response. The points addressed in the rebuttal are reasonable, and I recommend incorporating discussions on W-1 and W-3 into the revised version. However, my primary concern remains in the degree to which this work offers new insights to the community. For one, the brittleness of LLMs is widely observed, and many prior works have explored the integration of symbolic solvers or verifiers with LLMs in various components. Similarly, the limitations of FMs are well understood within communities like classical AI planning, expert systems, and symbolic reasoning etc. This work doesn't seem to introduce new perspectives, nor demonstrate the same depth as prior research that focuses solely on FMs. Overall, I will maintain my current score.

**Position:**

Yes

**Position In Title:**

Yes

**Related Work:**

4

**Strengths And Weaknesses:**

## Strengthes
- This manuscript provides a comprehensive overview of how FMs and LLMs can complement each other’s capabilities. Figure 2 is highly informative in illustrating the distinct roles that FMs and LLMs can play at different stages of a typical hybrid system.
- Hybrid approaches have got significant attention, and a discussion on this topic would be highly valuable for ICML.

## Weaknesses
While I appreciate the overall comprehensiveness of the manuscript, from the perspective of a position paper, it may lack the depth and strong stance necessary to challenge existing claims.

1. It is generally well understood that LLMs can streamline the process of establishing essential components for FMs. However, the key issue is what tradeoffs people have to make when employing hybrid approaches. For example, every formalism has its own limitations in terms of expressiveness. In other words, no single formalism can solve all types of problems. From the perspective of expressiveness and flexibility, are hybrid frameworks still advantageous over LLM-only systems? To provide a more robust position, I believe the manuscript should comment on these critical questions, rather than simply surveying existing research (sub-)fields.

2. Another key design choice in hybrid frameworks is whether LLMs should serve as the underlying solver or if an external symbolic solver should handle the final decision-making. In a position paper from last year [1], the authors argue that, for planning tasks, LLMs should be the underlying solver (in favor of expressiveness), with FMs used to act as guardrails for LLM outputs. I would like to see the authors take a position on this issue in the context of logical reasoning tasks.

[1] Kambhampati, Subbarao, Karthik Valmeekam, Lin Guan, Mudit Verma, Kaya Stechly, Siddhant Bhambri, Lucas Paul Saldyt, and Anil B. Murthy. "Position: LLMs can’t plan, but can help planning in LLM-modulo frameworks." ICML 2024.

3. Additionally, when discussing how LLMs can be leveraged to construct essential components for symbolic solvers, I suggest the authors have a deeper discussion on how the involvement of LLMs may compromise the conventional correctness guarantees provided by FMs. To my knowledge, verifiable correctness is extremely important in FMs.

Overall, while the manuscript does a good job of surveying relevant research areas, from the perspective of a position paper, it falls short in offering in-depth discussions on several critical open problems.

**Support:**

2

---

> ### Author Rebuttal · Authors · 2025-04-01
>
> We sincerely appreciate the reviewer’s thoughtful and insightful feedback and give the detailed clarifications and responses to the reviewer’s concerns and questions below.
>
> &nbsp;
>
> ### **Weakness**
>
> **W-1. Thank you for highlighting the trade-offs in integrating FMs with LLMs.**
>
> LLM-only systems inherently suffer from hallucination and reliability issues [1,2]. A key advantage of hybrid systems over LLM-only systems is that they can combine the generative flexibility of LLMs with the precision and rigor of formal methods. However, these benefits come at the cost of increased system complexity and challenges in seamless integration (different language mapping). Additionally, hybrid approaches may still be constrained by the expressive limits of the chosen formal methods. Therefore, we argue that such integration is more suitable for “downstream” tasks where correctness and reasoning are critical. However, for more flexible domains where the correctness or reasoning result doesn’t matter a lot, such as creative writing—an area highlighted by Reviewer zBM5—LLM-only systems, which prioritize generative flexibility, may be more effective.
>
> [1] LS Banerjee, A Agarwal, S Singla. LLMs Will Always Hallucinate, and We Need to Live With This. arXiv preprint arXiv:2409.05746 (2024).
> [2] Z Xu, S Jain, M Kankanhalli. Hallucination is inevitable: An innate limitation of large language models. arXiv preprint arXiv:2401.11817 (2024).
>
> On the other hand, the trade-off between the hybrid system and over FM-only system lies in adaptability/expressiveness vs. precision. FM-only systems excel in precision and rigor, providing exact specifications, proofs, and logical reasoning. They ensure correctness but often lack the flexibility and expressiveness to handle ambiguous, incomplete, or imprecise data that real-world problems often present. However, the hybrid system provides adaptability and flexibility to an extent, but may be inconsistent with the exact result during generation, which may need human labor to check the correctness manually. However, we argue that the specification-checking by human labor for the FM-only system is still an unavoidable process. Integrating LLMs does not introduce fundamentally new problems; rather, it shifts the nature of the challenges. It primarily reduces the manual effort involved (like in model specification construction) and enhances automation, but the core issues of ensuring correctness, rigor, and system design remain.
>
> We will add such a deep discussion in the revised version of the paper.
>
> &nbsp;
>
> **W-2. Thank you for highlighting the related paper [1].**
>
> Yes, we agree with the position that for the planning-related task, it would be better to use LLM to generate a potential solution, and FM serves as a guardrail. Indeed, such a position aligns with Section 3.3 (LLM for Theorem Proving) in our paper, where a complete proof can be regarded as a concrete plan in the planning task. As we mentioned in Section 3.3, LLMs help generate proof and FMs help to check the correctness of the proof. A more detailed example can be found in A.1.1, where we use Coq as a “guardrail” to check the proof construct by LLMs.
>
> Indeed, our position is that for traditional reasoning tasks, such as mathematical problems and deductive reasoning, formal methods should serve as the underlying solver due to their precision and rigorousness. However, for tasks with distinct planning characteristics, we argue that using LLMs to generate potential solutions, with formal methods serving as backend guardrails, represents a more effective design choice. This allows for the flexibility and expressiveness of LLMs while maintaining the necessary constraints and rigorousness provided by formal methods.
>
> We will add such a deep discussion in the revised version of the paper.
>
> &nbsp;
>
> **W-3. Thank you for pointing out the “correctness compromise” issue.**
>
> Indeed, the formal reasoning/checking process is not performed by the LLM; it remains the responsibility of FMs. The only non-automated aspect is the automatic generation of content (e.g., model/proof); the generated model/proof still needs to be verified by existing model checking and theorem proving algorithms, so this does not affect correctness. Therefore, as long as the verification, or formal analysis by FM tools, is passed successfully, the correctness of the system is not compromised.
>
> In the case of using LLMs for auto-formalization, we argue that the correctness will be guaranteed by human oversight. However, **as noted in our previous rebuttal to W-1**, this does not introduce a new problem. In conventional FMs, the correctness of the specification has always required verification by a human expert. Therefore, while LLMs can assist in automating parts of the process, the final assurance of correctness remains dependent on human intervention, as is the case with traditional FM approaches.
>
> We will add such a deep discussion in the revised version of the paper.

---

### Official Review · Reviewer_zBM5 · 2025-03-14

**Significance:** 3
**Argument Clarity:** 3
**Rating:** 3
**Confidence:** 4

**Questions:**

How can formal methods be effectively connected with problems that inherently require natural language reasoning?

**Discussion Potential:**

3

**Paper Summary:**

Large Language Models (LLMs) suffer from hallucination, making them unreliable. In contrast, formal methods provide mathematically rigorous verification but are limited by complexity and inefficiency. To build trustworthy AI, integrating LLMs with FMs is essential—combining LLMs’ adaptability with FMs’ formal guarantees to ensure both reliability and accessibility.

**Position:**

Yes

**Position In Title:**

Yes

**Related Work:**

2

**Strengths And Weaknesses:**

1. The proposed integration of LLMs and FMs is an intriguing approach worthy of discussion within the community. Formal methods serve as a fundamental pillar of rigorous reasoning in human knowledge.
2. Can formal methods handle all aspects of reasoning in daily life? For example, can they be applied to writing a novel or constructing a legal argument?

**Support:**

2

---

> ### Author Rebuttal · Authors · 2025-04-01
>
> We are sincerely grateful to the reviewer for acknowledging the strengths and weaknesses. Below, we give the response/answer for the concerns/questions raised.
>
> &nbsp;
>
> **Q1. Can formal methods handle all aspects of reasoning in daily life? For example, can they be applied to writing a novel or constructing a legal argument?**
>
> **A1.** Formal methods are powerful reasoning tools; however, they have inherent limitations when applied to complex, human-centered tasks such as novel writing or legal argument construction, particularly when starting from scratch. Generally, formal methods employ mathematical logic and rigorous proofs to ensure correctness, consistency, and reliability in domains such as software verification, hardware design, and mathematical theorem proving. However, novel writing involves emotions, aesthetics, and subjective interpretation—elements that are challenging to formalize. Similarly, although legal principles can be represented using formal languages, the interpretation of laws often requires moral reasoning and precedent-based judgment, which cannot be entirely encapsulated within rigid formal rules. Given these limitations, as insights given in this paper, a hybrid approach may be more effective. For instance, an oracle LLM can be employed to generate an initial legal argument, which can then be refined and evaluated using formal methods to assess its logical consistency and correctness in a semi-automated manner (given the legal principles formalized using formal languages).
>
> &nbsp;
>
> **Q2. How can formal methods be effectively connected with problems that inherently require natural language reasoning?**
>
> **A2.** A seamless connection between formal methods and natural language reasoning requires structured translation techniques (the mapping from natural language to formal language), AI-assisted formalization, and interactive verification tools. By leveraging both symbolic and statistical approaches, we argue that it is possible to enhance the applicability of formal methods in domains traditionally dominated by human judgment and linguistic ambiguity.
>
> Natural language problems are inherently ambiguous, while formal methods operate on precise semantics. So the **first step** is **formalization**—translating natural language into a formal representation (logic, constraints, automata, etc.). Once formalized, we can use **FM tools** (e.g., SMT solvers, model checkers) to perform **guaranteed reasoning**. Then the results can be **interpreted back into natural language** for human consumption or further LLM interaction.
>
> &nbsp;
>
> Once again, we appreciate the reviewer’s comments and hope our response addresses your questions.
>
>
> 2/2

---

### Official Review · Reviewer_McVf · 2025-03-14

**Significance:** 1
**Argument Clarity:** 2
**Rating:** 2
**Confidence:** 4

**Questions:**

1. How does the material from Sections 3 and 4 support the position of the paper? In particular, how is Section 3 related?

2. Why does Section 4.1 not belong in Section 3?

3. How is the position in this paper related to references [A,B,C]?

[A] Dalrymple, David, Joar Skalse, Yoshua Bengio, Stuart Russell, Max Tegmark, Sanjit Seshia, Steve Omohundro et al. "Towards guaranteed safe ai: A framework for ensuring robust and reliable ai systems." arXiv preprint arXiv:2405.06624 (2024).

[B] Wing, J. M. (2021). Trustworthy ai. Communications of the ACM, 64(10), 64-71.

[C] Seshia, S. A., Sadigh, D., & Sastry, S. S. (2022). Toward verified artificial intelligence. Communications of the ACM, 65(7), 46-55.

**Discussion Potential:**

2

**Paper Summary:**

The paper puts forth the position that a combination of LLMs and formal methods is necessary for building trustworthy AI agents. To support this position, the paper first presents several examples where LLMs have showed promised in making the use of formal methods easier. Next, the paper presents examples of using formal methods for aiding and analyzing LLMs in order to establish their trustworthiness.

## Update after rebuttal
Thank you for the response! I think the paper would benefit if the positions that (i) there should be a deep fusion of LLMs and FM, and (ii) behavior monitoring should be a feasible alternative to traditional verification methods, were stated more prominently and a effective case was made in favor of them. Although I have increased my score to weak reject, I continue to feel that the current paper's scope is too broad and generic.

**Position:**

Yes

**Position In Title:**

Yes

**Related Work:**

2

**Strengths And Weaknesses:**

### Strengths
1. Informing and reminding the ICML community about the powerful role that formal methods can play in improving trustworthiness of AI agents is a commendable exercise.

2. The model checking agent presented by the paper is interesting.

3. The paper can serve as a quick initial survey on ways in which LLMs have helped formal methods (Section 3).


### Weaknesses
1. The material in the paper does not present strong evidence in support of the position that formal methods and LLMs can be combined to build trustworthy AI agents. For instance, I find Section 3 to be completely unrelated to this position. Yes, LLMs can help apply formal methods but how does this help build trustworthy LLM/AI agents? While Section 4 does present some ways in which formal methods can help increase trust in AI agents, the presented techniques are either too specific or too weak. Section 4.1 in not even about increasing trust in AI agents and would perhaps fit better in Section 3 of the paper. Section 4.2 is a very specific approach to testing LLMs for hallucinations but its not clear what is the role of formal methods here. Section 4.3 proposes using formal runtime checks but does not say how such checks might be designed. Overall, I am left with the impression that paper surveys a few different ways in which LLMs and formal methods have been combined in the literature so far but it neither presents any evidence supporting the stated position nor a path towards the goal stated in the position.

2.  Although the PAT agent prototype and Section 4.2 seem to be new technical contributions, the writing of these sections is not very clear.

3. Similar positions have been articulated before [A, B, C].

[A] Dalrymple, David, Joar Skalse, Yoshua Bengio, Stuart Russell, Max Tegmark, Sanjit Seshia, Steve Omohundro et al. "Towards guaranteed safe ai: A framework for ensuring robust and reliable ai systems." arXiv preprint arXiv:2405.06624 (2024).

[B] Wing, J. M. (2021). Trustworthy ai. Communications of the ACM, 64(10), 64-71.

[C] Seshia, S. A., Sadigh, D., & Sastry, S. S. (2022). Toward verified artificial intelligence. Communications of the ACM, 65(7), 46-55.

**Support:**

1

---

> ### Author Rebuttal · Authors · 2025-04-01
>
> We sincerely appreciate the reviewer’s thoughtful feedback and engagement with our paper. Below, we address raised concerns/questions and clarify our position.
>
> ### **Clarification: Section 3**
>
> As we discussed in the paper, one of the major challenges in applying FMs in real-world applications is that FMs rely on strong expertise and manual work. If we want to have a deep fusion of LLMs and FMs in Section 4 to improve or certify the LLM agents’ trustworthiness, it is also essential to improve FMs’ scalability and automation, and an LLM is a promising way to improve FMs in these regards. Therefore, Section 3 (using LLMs to improve FMs) and Section 4 (using FMs to improve LLMs) go **hand-in-hand**–they **complement each other** and contribute towards the final goal of this paper. We foresee that future applications of LLM+FM will have the two aspects meshed together. But to structure the paper we separated them into two sections.
>
> We acknowledge that its connection to *trustworthy AI* was not explicit enough in the current draft. We will revise the manuscript to clarify how each category contributes to our position.
>
> ### **Weakness**
>
> **W-1.1**
>
> Section 3 explores the role of LLMs in supporting FMs, which, in turn, enhances the FM tools utilized in Section 4, as outlined in the clarification.
>
> **W-1.2**
>
> In Section 4, we explore various FMs that enhance LLM trustworthiness, providing research status and solutions to specific challenges. These methods are crucial for developing trustworthy LLM agents, and we will clarify how they can be extended and combined into a training-free pipeline.
> - Regarding Section 4.1, it is difficult to make a strict "X for Y" distinction. While it involves technical challenges related to autoformalization, its main objective is to use SMT solvers to enhance LLM trustworthiness, aligning it more with Section 4 (using FMs to improve LLMs). Section 3.1, by contrast, focuses exclusively on autoformalization.
> - Section 4.2 applies logical reasoning (a type of FMs) to create a comprehensive benchmark for evaluating LLM performance via testing.
> - Section 4.3 discusses the intrinsic challenges in LLM verification and **gives our position** that behavior monitoring should be a feasible alternative to traditional verification methods, along with future research challenges in this direction.
>
> **W-2**
>
> We appreciate the reviewer’s feedback on Section 3.2 and will refine it to better clarify the framework’s motivation, workflow, and unique aspects in the revised version.
>
> We will clarify that PAT is a low-resource formal language with limited training data, making integration with LLMs challenging, but also showing strong potential for generalizability to other FM tools. Unlike LLM applications that tolerate loose semantics, our model-checking agent ensures strict formal guarantees, with all generated outputs provably correct via formal verification.
>
> We will emphasize that our contribution is more than a proof-of-concept; it is a functional framework that leverages LLMs’ strengths–such as natural language understanding and code synthesis–within a formal setting, incorporating iterative correctness checking and refinement. The agent provides a user-friendly interface, enabling users unfamiliar with FMs to generate formally verified code, thereby promoting broader adoption of trustworthy AI practices.
>
> **W-3. Comparison against Papers [A,B,C]**
>
> We appreciate the reviewer for highlighting these relevant works. We will add them to the related work and include the following discussion.
>
> [A,B,C] focus on trustworthiness in general AI systems, particularly in domains like robotics and cyber-physical systems. They primarily use traditional FMs like model checking and verification, emphasizing challenges such as world modeling, specification design, and verifier implementation. These works adopt a unidirectional approach, applying FMs to AI systems to verify their correctness.
>
> In contrast, our paper explores the fusion of LLMs with FMs to tackle the unique challenges of ensuring LLM-driven agents' trustworthiness. We emphasize the synergistic relationship between LLMs and **a broader range** of FMs, including specification, model checking, verification, theorem proving, and automated reasoning. Unlike [A,B,C], which focus on verifying a mathematical model against a specification, we explore how LLMs and FMs can enhance each other’s capabilities, ultimately improving or certifying the trustworthiness of LLM-driven agents.
>
> Moreover, [A,B,C] use a **unidirectional approach** to apply FMs to AI systems, while our paper gives a **bidirectional** integration, highlighting how LLMs can enhance FMs (Section 3) to improve FMs’ efficiency and adaptability, and how FMs can help certify LLM-driven agents' trustworthiness (Section 4).
>
> ### **Questions**
> See rebuttals to clarification and weakness.
>
> **Once again, thank you for your thoughtful comments. We will refine the paper for greater clarity and focus.**

---

### Decision · Program_Chairs · 2025-04-30

**Decision:**

Accept (poster)

**Comment:**

This is a nice paper that outlines the pitfalls of LLMs and their trustworthiness, and proposes solution of hybrid architecture that combines LLMs with FMs (Formal Methods). This paper serves as a nice survey and introduction to the area of FM and at the same time proposes thought provoking concepts. In particular, Section 3.2. LLM for Model Checking is quite impressive.

Note: The ICML footer is missing.  Authors should ensure they are using the ICML 2025 style files.